# RAAS: A Convergent Player in Ischemic Heart Failure and Cancer

**DOI:** 10.3390/ijms22137106

**Published:** 2021-07-01

**Authors:** Texali C. Garcia-Garduño, Jorge R. Padilla-Gutierrez, Diego Cambrón-Mora, Yeminia Valle

**Affiliations:** 1Centro Universitario de Ciencias de la Salud, Instituto de Investigación en Ciencias Biomédicas, Universidad de Guadalajara, Guadalajara 44340, Mexico; garciatexali@gmail.com (T.C.G.-G.); imey_99@yahoo.com (J.R.P.-G.); 2Doctorado en Genética Humana, Departamento de Biología Molecular y Genómica, Universidad de Guadalajara, Guadalajara 44340, Mexico; 3Doctorado en Biología Molecular, Departamento de Biología Molecular y Genómica, Universidad de Guadalajara, Guadalajara 44340, Mexico; diegocambron21@gmail.com

**Keywords:** heart failure, myocardial infarction, cancer, renin-angiotensin-aldosterone system

## Abstract

The current global prevalence of heart failure is estimated at 64.34 million cases, and it is expected to increase in the coming years, especially in countries with a medium-low sociodemographic index where the prevalence of risk factors is increasing alarmingly. Heart failure is associated with many comorbidities and among them, cancer has stood out as a contributor of death in these patients. This connection points out new challenges both in the context of the pathophysiological mechanisms involved, as well as in the quality of life of affected individuals. A hallmark of heart failure is chronic activation of the renin-angiotensin-aldosterone system, especially marked by a systemic increase in levels of angiotensin-II, a peptide with pleiotropic activities. Drugs that target the renin-angiotensin-aldosterone system have shown promising results both in the prevention of secondary cardiovascular events in myocardial infarction and heart failure, including a lower risk of certain cancers in these patients, as well as in current cancer therapies; therefore, understanding the mechanisms involved in this complex relationship will provide tools for a better diagnosis and treatment and to improve the prognosis and quality of life of people suffering from these two deadly diseases.

## 1. Introduction

Heart failure (HF) and cancer are the main public health issues in industrialized countries and are increasing in prevalence, especially, in the ageing population [1]. These two diseases were thought to be independent; however, new research has revealed that cancer and HF frequently coexist in the same patient. Furthermore, as cancer-specific mortality decreases and the surviving population gets older, the overlap between cardiac disease and cancer patients is growing. As a result, the discipline of cardio-oncology has primarily focused on the adverse effects of anti-cancer therapy. HF is one of the most serious consequences of cardiotoxic cancer therapy [2,3].

In recent years, various epidemiological studies have reported a new scenario: patients with HF have an increased risk of cancer compared to the general population [4,5,6,7,8], and this new connection represents a challenge for the field of cardio-oncology. Establishing the pathophysiological and molecular mechanisms that converge in these two diseases is of vital importance since they represent two of the main causes of morbidity and mortality worldwide. In addition, it has been shown that mortality increases substantially in the presence of both entities [9].

These two pathologies share a variety of cardiovascular risk factors as well as pathophysiological mechanisms. There have been two hypotheses proposed about the shared pathophysiological mechanisms: a constant inflammatory environment and a chronic neurohumoral activation [10]. Neurohumoral response includes the renin-angiotensin-aldosterone system (RAAS), which can be chronically stimulated, representing a hallmark of HF [11,12], and it has also been established that RAAS is frequently altered in a variety of cancer types, which is associated with a poor prognosis [13,14]. Moreover, RAAS can exert a chronic proinflammatory response mediating a substrate for both HF and cancer development, and our intention here is to complement this knowledge with recent scientific contributions from the field of cardio-oncology.

## 2. Clinical and Preclinical Evidence of Heart Failure Triggering Cancer

The American College of Cardiology defines HF as “a complex clinical syndrome that results from any structural or functional alteration of ventricular filling or blood ejection and, the most common manifestations are dyspnea and fatigue which can restrict exercise tolerance, and fluid retention that can contribute to pulmonary and/or splanchnic congestion and/or peripheral edema” [15]. The current world prevalence of HF is estimated at 64.34 million cases (8.52 out of every 1000 habitants) with an increase of 3.1% in the last decade, where the regions with a medium–low sociodemographic index (SDI) represent 43.4–46.5% worldwide, with an estimated prevalence of more than 50% by the year 2030 [16]. The etiology of HF encompasses various conditions, but coronary artery disease (CHD, also referred to as atherosclerotic cardiovascular disease or ischemic heart disease) remains the most common cause of HF [17,18]. CHD manifests clinically as myocardial infarction and ischemic cardiomyopathy [19], but in particular, acute myocardial infarction (MI) is among the most frequent underlying causes [20,21,22]. As previous reports indicate “heart failure can be viewed as the chronic stage of any disease leading to cardiac functional impairment”. It is noteworthy that multiple causes frequently coexist, and most (comorbidities) do not arise separate from HF, but rather share a set of risk factors [23,24].

The American Cancer Society defines cancer as a group of diseases characterized by uncontrolled growth and the spread of abnormal cells that can cause death if not treated [25]. In 2020, globally, there were 19.3 million new cases, with female breast cancer being the most commonly diagnosed type, followed by lung and colorectal cancer; and nearly 10 million deaths with lung cancer leading the mortality chart, followed by colorectal and liver cancer [26]. It has been established that cancer cells must possess certain distinctive and complementary capacities to survive, proliferate and spread, among which we can list the maintenance of proliferative signaling, the evasion of growth suppressors, the avoidance of cell death, induction of angiogenesis, reprogramming of energy metabolism, inflammation, activation of cell migration, invasion and metastasis, evasion of the immune response, replicative immortality, and genome instability [13,14,27].

There are quite a few reports in the literature on the association of cancer as a trigger for HF; however, the idea that it is HF that causes a complex pathology such as cancer had not been considered until a few years ago, and has taken on a relevant role, as potential associations have been reported on the basis of epidemiological, systematic, and even experimental studies. Such is the case of five observational epidemiological studies, which reported a positive association in the increased risk of cancer in individuals with HF, even up to two times more than in the control group [4,5,6,7,8]. The major findings of these studies are summarized in Table 1. As a result of these epidemiological studies, the incidence of cancer in this population has been estimated to be around 18.9–33.7/1000 inhabitants per year [28]. Nevertheless, Selvaraj et al., in a study that included a large number of individuals, did not observe this association [29]. It should be noted that in this study, only males were included, and since it was observed that sex influences HF, the study presents an important bias [30]. Other notable situations are that Selvaraj et al. ruled out individuals who had previously reported MI, an important contributor to HF worldwide [20,21,22], and the fact that the exposure of interest (HF) was a self-reported diagnosis, while cancer outcomes were first ascertained by annual self-report (this is because the study participants were physicians), instead of being assigned according ICD-9-CM diagnosis code 428 [31].

Recently, a meta-analysis was carried out to explore cancer morbidity and co-mortality in adults with HF in a large sample by Zhang et al. in which the main conclusions were, on the one hand, that HF increased the risk of cancer and, second, that cancer associated with HF increased mortality. Comorbidity analysis showed an increased risk of cancer within the HF group that had a previous MI [9]. In addition to this, experimental studies in animal models that have studied the association of cancer with HF start from the induction of MI [32,33], so it could be inferred that this is a crucial factor. It is also worth noting that in the observational study carried out by Hasin et al. whose group of study were survivors of a first MI with and without subsequent HF, when analyzing the incidence of cancer between groups, it was observed that this was initially similar, but diverged after 1.5 years of follow-up, with higher rates of cancer among HF patients [7]. However, the incidence of cancer concerning the group of patients with MI but without HF was present and even showed an increasing trend, but it was not as noticeable as in the group with HF. Another situation that could affect the findings is the underdiagnosis of MI, because the aforementioned studies were carried out before the introduction of high-sensitivity troponin (hs-cTn), which as its name indicates, is a much more specific biomarker of myocardial damage than previously used assays, since this assay detects the presence of cardiac troponin (cTn) earlier than previous tests, leading to improved sensitivity for the diagnosis of MI [34].

Additionally, Koelwyn et al. recently reported that myocardial infarction accelerates breast cancer growth and cancer-specific mortality in mice and humans without the overt clinical presentation of HF, at least in the murine model [32]. It is worth mentioning here that both MI and HF share a similar pathophysiological substrate, and MI cannot be ruled out as an intervening factor with as much weight as HF has been shown to be; however, since the moment in which HF occurs after myocardial infarction is often not clearly defined in research studies, it is difficult to analyze [21]. As the pathological environment that gives rise to HF through MI may be different from other etiologies, it should be noted that this document will point out what has been described HF caused by MI.

## 3. Renin-Angiotensin-Aldosterone System Linking Ischemic Heart Failure and Cancer

There is evidence that the RAAS is involved in most of the tumorigenic characteristics described above, and due to the chronic activation of RAAS in HF, it has been proposed that a failing heart may be closely related to the development of cancer. In this review, we will discuss recent studies that highlight the role of RAAS components as an axis of crucial importance in the pathophysiology of HF and as well as evidence of the dysregulation of its components in the development of cancer to highlight the points where these two entities that were previously considered independent could now converge.

There are two major pathways in the RAAS: classical and non-classical pathways. In the classical RAAS, the effector peptide is angiotensin-II (AngII), which is produced from its hepatic precursor, angiotensinogen, which is catabolized by the enzyme renin, giving rise to angiotensin-I (AngI) in turn, which is a substrate for the angiotensin-converting enzyme (ACE) producing angiotensin-II. The functional effects of AngII in the classical RAAS are largely mediated by the type 1 angiotensin-II receptor (AT1R) and the type 2 receptor (AT2R) [12,35]. AT1R activation increases aldosterone, an important player in the regulation of electrolyte balance [36], but AT1R activation also has many other effects (described later). Signaling mediated by the AT2R is associated with antifibrotic functions and even with anti-inflammatory effects in HF [37,38,39], while in cancer, this axis has antiproliferative, antiangiogenic, and pro-apoptotic effects [40,41]. However, there are also conflicting reports suggesting possible tumor type-specific differences [36,40].

In the non-classical RAAS, the homologue of ACE, angiotensin-converting enzyme 2 (ACE2) cleaves AngI into a nonapeptide, Ang 1-9 and AngII into a heptapeptide, Ang 1-7. Additionally, AngII can be also converted to Ang 2-8 (AngIII) by aminopeptidase A, and exerts its effects by binding to AT1R. Aminopeptidase N converts AngIII to Ang 3-8 (AngIV) and can act through the angiotensin 4 receptor (AT4R) [12]. Ang 1-9 can activate AT2R, and Ang 1-7 can bind to the proto-oncogene Mas receptor (MasR). Interestingly, every one of these components has been demonstrated to counteract the actions of the classical RAAS [12,42]. Signaling mediated by the ACE2/Ang 1-7/MasR axis has been shown to have a protective role in the development of myocardial remodeling post-MI in an animal model [43], but it is also associated with antifibrotic and anti-inflammatory effects [44,45]. Moreover, AngIV/AT4R signaling has a cardioprotective role, acting as a counterpart of Ang II-mediated inflammation and myocardial fibrosis in rat model [46]. In cancer, MasR has been documented to reduce abnormal angiogenesis, inflammation and cell proliferation by the local decrease of Ang II levels or AT1 receptor blockade associated with high concentrations of Ang(1-7) at the tumor site [47]. Even so, as the AT1R continues to be crucial in mediating physiological and pathophysiological effects of AngII [48], in this review, we are going to focus in the classical AT1R/AngII RAAS axis.

AngII overproduction is linked to the development of chronic illnesses; in fact, a chronic activation of RAAS is a hallmark of HF, especially marked by a systemic increase in levels of AngII [10,49], and to better understand how RAAS is implied in both of these diseases, first we must consider the involved pathophysiology from MI to HF and then later to cancer.

Cardiomyocyte necrosis in the infarcted myocardium activates the innate immune response, triggering an inflammatory response. The release of danger signals from dying cells induces the secretion of cytokines, like chemokines and adhesion molecules, to allow the recruitment and infiltration of leukocytes, mainly monocytes, into the infarcted area, where they exert a “reparative” response, phagocytosing the cellular debris, while stimulating repair pathways by secreting pro-inflammatory cytokines. To supply the appropriate number of immune cells, a release of stem cells and hematopoietic progenitors from the niches of the bone marrow occurs; these cells then migrate to the spleen and, ultimately, increase the production of immune cells, which in turn mediates an effective inflammatory response (Figure 1) [50,51]. The modulation of inflammation in this repair phase includes fibroblast activation and healing mediated by the neurohumoral response. RAAS, which is part of the neurohumoral response, is activated by renal hypoperfusion and sympathetic activation as compensatory mechanisms after a myocardial injury [11]. In these processes, RAAS actively participates mainly through the AngII effector peptide. Indeed, various components of RAAS, including angiotensinogen, AngII, ACE, AT1R, and AT2R, have been reported to be expressed in a variety of immune cells [52], as well as in bone marrow cells [53]. Shortly after myocardial injury, an increase in AngII concentration occurs, which induces an accumulation, differentiation, and the exit of hematopoietic stem/precursor cells (HPSC) from the bone marrow to contribute to splenic myelopoiesis, supplying up to 50% of the leukocytes to the infarcted area [54], and through the phosphorylation of nuclear factor-kappa B (NF-kB), the binding of AngII to its AT1R receptor induces a pro-inflammatory response mediated by tumor necrosis factor-alpha (TNF-α) or interleukin-1 beta (IL1ß), which in turn are drivers of inflammation [55]. Nevertheless, when neurohumoral response becomes chronic, it leads to an excessive loss of cardiomyocytes, an exacerbated inflammatory response, and the healing and adverse remodeling of the infarcted ventricle, which ultimately underlies HF [56,57,58]. This dysfunctional environment has been proposed to trigger the secretion of several factors into the circulation that can be synthesized in various cell types surrounding the heart and its own cell components, including cardiomyocytes, fibroblasts, smooth muscle (aortic or blood-derived progenitors), and vascular endothelial cells (Figure 1) [33,59].

In this scenario, one of these secreted or leading factors can be components of RAAS, especially, AngII. This statement is based on several facts. On one hand, the activation of AngII/AT1R axis is generally associated with the pathophysiological appearance in HF, and it has also been established that RAAS is frequently altered in a variety of cancer types, which in turn is associated with a poor prognosis [13]. It should be considered that the pathological effects observed in these diseases are mainly associated with the AngII/ATR1 axis [45]. Thus, AT1R signaling increases aldosterone levels and blood pressure, induces vasoconstriction, cardiac hypertrophy, fibrosis, inflammation, and reactive oxygen species (ROS) production, while decreasing nitric oxide (NO) production, among other effects [12,42]. In the cancer scenario, AT1R activation by AngII favors cell proliferation, inhibits apoptosis, and promotes adhesion molecule expression, the interaction of monocytes with endothelial cells (EC), the infiltration of inflammatory cells, and the generation of pro-inflammatory cytokines, enabling the establishment of the inflammatory microenvironment, which is a pivotal state for the subsistence of neoplastic cells [47].

On another topic, it has previously been proposed that the initial immune response against a neoplasm is the result of the presence of an acute tissue injury that has generated a chronic infiltration of various myeloid cells, triggering a state of chronic inflammation in the tissue environment because the initial acute inflammatory response did not resolve [60]. Along the same lines, RAAS is a driver of tumorigenesis, linking with HF through immune and inflammatory responses. The involvement of immune cells both in the acute response after MI, the progression towards HF, and in the tumor microenvironment (TME) is a well-established notion. In the TME, immune cells intervene in various stages, mainly due to their infiltration into tumors and their differentiation into tumor-associated macrophages (TAM) [61], which are important components of the infiltration of most tumors and are derived mainly from circulating monocytes and which are attracted to the tumor by chemokines. In such a tumorigenic microenvironment, TAMs can stimulate tumor cell proliferation, promote angiogenesis, and favor invasion and metastasis [62]. In TME, they are frequently located surrounding blood vessels, where they secrete vascular endothelial growth factor (VEGF) and induce new blood vessel formation (angiogenesis). TAMs are the major immunoregulatory cells, and result in immune suppression in TME [63]. It should be noted that in the post-MI phase, monocytes predominate [32,57]. In MI, monocytes secrete angiogenic mediators, such as VEGF and the fibrogenic mediator transforming growth factor-beta (TGF-β), where their function is to promote the repair of infarcted tissue and cardiac fibrosis [57], so these HF-immune mediators can act as a substrate for the development of cancer cells by modulating a favorable microenvironment for its development (Figure 2).

Additionally, it was reported that systemic changes induced by the tumor influence the phenotype of circulating monocytes (such as the acquisition of immunosuppressive activity and a decreased responsiveness to inflammatory stimuli) before their infiltration into the tumor environment [61]. In particular, the inflammatory Ly6C^high^ monocyte subset is efficiently recruited towards tumors and provides mediators that stimulate cancer-associated inflammation and angiogenesis [54,61]. Consistent with the above, Koelwyn et al. reported that MI accelerates breast cancer growth and cancer-specific mortality in mice and humans. In a murine model, there was an increase in the levels of circulating Ly6C^high^ monocytes that were epigenetically reprogrammed in the spleen towards an immunosuppressive phenotype that was maintained in the tumor, as well as in the blood circulation, and additionally demonstrated that the depletion of these cells abolished MI-induced tumor growth [32]. AngII plays a relevant role in macrophage-mediated chronic inflammation by modulating this macrophage amplification program, since according to Retamozo et al. the overproduction of AngII increased macrophage progenitors in the spleen, allowing the extramedullary tissue to supply new macrophages associated with tumors throughout cancer progression in a tumor-bearing animal model. In contrast, blocking AngII production prevented the amplification of macrophage progenitors [54]. In this context, RAAS, and in particular AngII, could be a key point in the convergence of HF and cancer pathophysiology.

That is why it has been stated that chronic inflammation is a point of convergence between HF and cancer; because the former is characterized by chronic inflammation, it directly influences the risk of cancer development in patients with HF, since inflammation is an established component of carcinogenesis. Evidence suggests that chronic inflammation is responsible for up to 25% of all cancers [64]. On this occasion, we want to highlight the role of endothelial dysfunction as a pathophysiological substrate in the development of MI and HF, which in turn can generate an environment conducive to cancer progression.

The presence of cardiovascular risk factors, among them obesity, diabetes, and hypertension, directly affects the endothelium which is composed by EC. These cells line the inside of all major and minor vessels and serve as the first point of contact between the lumen and other tissues and regulate vascular tone, stiffness, inflammation, thrombotic potential in both health and illness (Figure 2). Cardiovascular risk factors mediate their detrimental effects on the vessel wall in part via enhanced activity of RAAS and increased release of vasoactive agents including Ang II as well as paracrine and circulating factors that regulate the generation and activity of endothelium-derived vasoactive and growth factors, adhesion molecules that mediate leucocyte-EC interaction, and blood coagulation regulators [65,66]. The endothelium in a healthy vasculature is anti-inflammatory, anti-thrombotic, and promotes vasodilation but, on the contrary, when the endothelium becomes dysfunctional, it is characterized by a pro-inflammatory and pro-thrombotic state [67,68] (Figure 2). It is worth noting that the exact mechanism by which a normally functioning endothelium becomes dysfunctional remain unknown. However, it has been reported that the endothelium is a prime site for the effects of cardiovascular risk factors; thus, endothelial function can be seen as an integrated index and sensitive measure of cardiovascular disease risk, since it reflects the cumulative contribution of various risk variables associated with inflammation and oxidative stress and given the similar pathological mechanisms that underpin cancer and cardiovascular disease [69]; thus, this is an elemental cellular component that can intermediate the transition between HF and cancer (Figure 2).

From a cancer perspective, dysfunctional ECs can promote pro-inflammatory signaling that is associated with characteristics that favor cancer progression, while in non-pathological conditions, it has been reported that ECs mitigate tumor invasiveness and metastasis [70]. Using in vitro models of dysfunctionally activated ECs, Franses et al. observed that resting EC constructs exhibited moderate inflammatory activity and could inhibit the proliferation and invasion of cancer cells. In contrast, “dysfunctional” ECs favored spontaneous metastasis in adjacent tumors through an aberrant expression of pro-inflammatory cytokines, extracellular matrix, alterations in the leukocyte adhesion process, increasing the expression of vascular cell adhesion molecule-1 (VCAM-1) and abnormal responses to oxidative stress, which are pathological stimuli present both in atherosclerotic lesions, precursors of MI and HF, as well as in the tumor environment (Figure 3) [66,70]. Molitor et al. provided evidence that the AngII/AT1R axis favors the migration and infiltration of myeloid cells towards the aortic walls, inducing endothelial dysfunction. Notably, AT1R blockade with telmisartan attenuated vascular infiltration of immune cells, reducing oxidative stress, and improved endothelial dysfunction [71]. Furthermore, in a subsequent study, they tested ACE inhibition in an HF model after MI, where they observed a decrease in systemic inflammation accompanied by a reduction in vascular infiltration of inflammatory myeloid cells and a decrease in the ROS levels nicotinamide adenine dinucleotide phosphate oxidase (NADPH oxidase) produced [55].

ECs are essential in the tumor microenvironment, as they can also express and use components of the RAAS signaling pathway to promote tumor growth, enhance angiogenesis, and promote metastasis [13,72]. It should be noted that adhesion molecules play a crucial role in these processes, since they allow the union and transendothelial migration, in this case, of tumor cells or TAMs. AngII signaling activates TNF-α and NF-κB and upregulates pro-inflammatory endothelial chemokines [66,73]. AngII has been reported to be able to promote VCAM-1 expression and enhance adhesion, growth, angiogenesis, and the inflammatory microenvironment through AT1R in hepatocellular carcinoma [74]. During HF, an increase in VCAM-1 expression was observed in response to AngII stimulation, and this is associated with endothelial dysfunction as well [71,75].

Angiogenesis is involved in myocardial healing. It has been reported to promote tumor progression, depending first on supplying oxygen and nutrients and later, generating a pathway for its metastasis [14]. In ECs, AT1R-mediated AngII generates a pro-angiogenic response mediated by VEGF, a crucial stimulator of pathological vessel formation (Figure 3) [76]. Therefore, this represents another mechanism in which HF could influence the environment that leads to cancer progression. Thus, the study of endothelial dysfunction mechanisms is crucial to preventing the recurrence of serious secondary events in patients who have suffered HF, including the development of cancer.

## 4. Role of Drug Therapy with RAAS Inhibitors in Heart Failure and Cancer

Angiotensin-converting enzyme inhibitors (ACEIs) and angiotensin receptor blockers (ARBs) are part of the basic pharmacological treatment after MI (besides reperfusion), and its administration has been reported to reduce mortality in the short and long term in addition to reducing the risk of HF [77,78,79]. Additionally, recent data suggest that these drugs also possess anti-inflammatory and anticancer characteristics [80,81], which reinforces the convergent role of the RAAS system in HF and cancer, because their molecular inhibitors play an important role in development, migration, recurrence, and resistance to antineoplastic drugs [81].

To denote the role that RAAS plays in HF and cancer, we want to highlight the therapeutic properties that RAAS inhibitors possess in HF (Table 2) and the findings that these inhibitors have shown in cancer preclinical research (Table 3).

There is also clinical evidence regarding the use of RAAS inhibitors in a variety of cancer types. Song et al. reported that the use of ACEIs reduced the risk of cancer recurrence and mortality by 40% and 25% when ARB was administered, showing significant effects for colorectal, urinary, pancreatic, and prostate cancer, but not in hepatocellular and breast cancer [95], and a positive association has also been noted for renal [96,97] and intestinal cancer [97]. However, the lack of association of the effect of these inhibitors on some types of cancer has also been reported [95], which could reflect the complex biology of cancer, including the specific characteristics of each tumor, since it has been described that tumors at different sites have their own intrinsic and extrinsic mechanisms, as well as concerning the presence of cardiovascular risk factors. In the case of RAAS components, for example, AT1R has been reported to be more expressed in breast, larynx, pancreas, and choriocarcinoma cell lines [81]. However, clinical studies are also required in the appropriate population to know the true role of these inhibitors in the development of cancer in patients with pre-existing cardiac conditions.

A relevant aspect of ACEI/ARB drugs was recently reported by Fang et al., who conducted a meta-analysis of randomized controlled trials and observed that pharmacological administration of these two types of RAAS inhibitors attenuated cardiac dysfunction produced by chemotherapy [98]. Nonetheless, experimental studies concerning the molecular mechanisms for the potential benefits of RAAS inhibitors with respect to cardiotoxicity are rare, for this reason, future research is needed to evaluate whether the benefit of using ACEI/ARBs in this group of patients is clinically relevant.

Additionally, promising results have been reported for the use of ACEI and ARBs in the treatment of other inflammatory diseases such as atherosclerosis, arthritis, and colitis [80], so it is noteworthy that their anti-inflammatory effect could be a mechanism involved in reducing cancer mortality.

**Table 3 ijms-22-07106-t003:** Role of RAAs inhibitor in cancer preclinical research.

RAAS Inhibitor	Findings
**Angiotensin converting enzyme inhibitors (ACEI)**	
Captopril	Inhibits tumor growth in a gastric cancer model and suppresses the angiogenesis of the tumor by decreasing the expression of vascular endothelial growth factor (VEGF) and matrix metalloproteinase (MMP)-7 in a mouse model with human gastric cancer [99]. Attenuates cell migration in a breast cancer model [100]. Inhibits cell growth, decreases *c-myc* expression, and increases apoptosis on leukemic cell lines [101].
Enalapril	Inhibits tumor progression and reduces number of tumor-associated macrophages (TAMs) [54].
Perindopril	Can inhibit the tumor growth in gastric cancer model and suppress the angiogenesis of the tumor by decreasing the expression of VEGF and MMP-7 in a mouse model with human gastric cancer [99].
Ramipril	Decreases systemic inflammation [55].
Trandolapril	Inhibits cell growth, decreases *c-myc* expression, and increases apoptosis in leukemic cell lines [101].
**Angiotensin II type 1 receptor blockers (ARBs)**	
Telmisartan	Inhibits cell proliferation and tumor growth of esophageal squamous cell carcinoma by inducing s-phase cell cycle arrest [102].
Candesartan	Prevents bladder cancer growth in a mouse model by inhibiting angiogenesis, and combined treatment with candesartan and paclitaxel enhances paclitaxel-induced cytotoxicity [103]. Candesartan treatment significantly sensitizes human lung adenocarcinoma cells to tumor necrosis factor-related apoptosis-inducing ligand-mediated apoptosis [104].
Losartan	Can inhibit the tumor growth in gastric cancer model and suppress the angiogenesis of the tumor decreasing the expressions of VEGF [99]. Can exert anti-metastatic activity by inhibiting chemokine receptor type 2 (CCR2) signaling and suppressing monocyte recruitment in a mouse model with tumors and indirectly as anti-inflammatory effect and independently of AT1R [105]. Ameliorates angiogenesis, inflammation and the induction of oxidative stress via type-1 angiotensin-II receptor (AT1R) in a murine model of lung metastasis of colorectal cancer [106]. Inhibits cell growth, decreases *c-myc* expression and increases apoptosis in leukemic cell lines [101].
Valsartan	Can inhibit the tumor growth in gastric cancer model and suppress the angiogenesis of the tumor, decreasing the expressions of VEGF [99].
**Aldosterone antagonists**	
Spironolactone	Inhibits cancerous cell growth and is highly toxic for cancer stem cells; impairs DNA-double-strand breaks repair and induces apoptosis in cancer cells and cancer stem cells (CSCs) while sparing healthy cells. In vivo, this treatment reduces the size and CSC content of tumors [107].

## 5. Future Directions

The diagnosis of cancer in patients with HF has notable prognostic and therapeutic repercussions: on the one hand, the mortality associated with the coexistence of these diseases increases considerably, as concluded in observational studies; additionally, it has been reported that a great variety of antineoplastic drugs are cardiotoxic [108,109,110], so their administration in patients with pre-existing HF should be considered with caution. Therefore, the search for new therapeutic strategies with a lower cardiotoxic effect or the search for adjuvant drugs with a cardioprotective effect is an opportunity for future research in an era where life expectancy has increased significantly, leading to an increase in the number of elderly patients, so the coexistence of these two diseases could be more common in the coming years.

Another intriguing situation arose from the results of contemporary registries, revealing that, at minimum, 1 in 10 patients with HF with reduced ejection fraction also has a malignant tumor at the first observation or is diagnosed with and dies from cancer during follow-up [111]. It must be mentioned that most studies exploring this cardio-oncologic situation excluded patients with this characteristic. Therefore, some questions emerge: what is happening in such individuals? Did cancer and HF develop at the same time and due to the same substrate? In addition, what are the specific types of cancer?

Traditional MI/HF treatment includes the prescription of ACEIs and ARBs, drugs that have been used as a strategy to preserve the left ventricular ejection fraction, in subjects undergoing chemotherapy treatment [98,109]. Additionally, a reduction in the risk of cancer recurrence and mortality by up to 40% has been shown, in particular for colorectal, urinary, pancreatic and prostate cancers [95]. The reduction in the risk of cancer was observed in patients with a treatment of ACEI or ARBs, and not with the use of other antihypertensives; this suggests that the biological mechanism involved is not associated with the reduction of blood pressure [81], generating an area of study of the specific mechanisms by which this effect takes place. Additionally, it has been documented that RAAS AngII/AT1R axis inhibitors, apart from decreasing the levels of AngII, could prevent myocardial remodeling after MI through an increase of the ACE2/Ang 1–7/MasR axis, as shown in an animal model [43]. Together, this information provides a new approach for developing drugs that could mediate this dual effect.

HF is a complex, multi-stage disease that occurs when subclinical conditions, such as endothelial dysfunction and atherosclerosis, have progressed to manifest acutely. Endothelial cells are the most common type of non-myocyte cell in the heart [68] and have been defined as a “cardiovascular risk barometer” due to their role in pathologies such as those described in this review. Endothelial dysfunction leads to the predisposition of a series of pathological conditions such as MI, HF, and cancer; and again, RAAS inhibitors could mitigate this condition, since it has been reported that in addition to their protective effects on cardiac remodeling and blood pressure control, they exert an effect on systemic inflammation and prevent endothelial dysfunction associated with myeloid cell infiltration in the aortic wall [55]. In the tumor environment, ECs are capable of expressing and using components of the RAAS signaling pathway to promote tumor growth, enhance angiogenesis and promote metastasis by providing components such as VEGF, a pro-angiogenic agent, and adhesion molecules such as VCAM1. Therefore, the molecules secreted by the endothelium have been studied as a therapeutic target in cancer. However, the mechanisms involved in the transformation of an integral endothelium towards a dysfunctional endothelium are not completely known. This is why there is a need to further investigate the properties of this cellular component in order to develop more specific therapeutic strategies, as well as clinical tools and preventive measures that allow the identification of not-so-serious clinical conditions and thus anticipate the presence of complex pathologies such as HF and cancer.

Another interesting area of research is immunotherapy complemented with RAAS inhibitors. Immune checkpoint blockade aims to enhance the immune response of the host to the different progression stages of tumors; this has shown fewer off-target effects compared with the chemotherapy approach or other treatment therapies that destroy cancer cells directly [112]. The immune checkpoint antibodies against cytotoxic-T-lymphocyte-associated protein 4 (CTLA-4), programmed cell death protein 1 (PD-1), or together have been shown to be effective treatment options for a range of malignancies [113,114]. Despite this fact, only a small percentage of patients have long-lasting responses, and almost 70% of patients do not respond to single-agent regimens for most types of malignancies. Immune resistance to these medications is caused by a variety of mechanisms [115,116], including the intratumor immunosuppressive microenvironment, which is the most critical element facilitating tumor cell immune evasion in vivo [63]. Interestingly, Xie et al. evaluated the effect of local AngII in the TME. They demonstrated that this peptide plays an important role in the formation of a tumor immunosuppressive microenvironment; in addition, they reported that the blockage of AngII effector functions greatly improved tumor sensitivity to checkpoint immunotherapy in a murine animal model. Thus, the use of an AngII signal blocker in association with a checkpoint antagonist could be a promising approach to improving tumor responses to current anti-checkpoint immunotherapy [115]. Moreover, next-generation immune-modulating therapy considers not only the reinvigoration of exhausted or suppressed T cells in the TME as anti-checkpoint immunotherapy does, but also indicates that TME contains various other immune cell subsets that interact to determine the fate of cytotoxic T cells [117]. In this preamble, it was reported that a variety of the classical RAAS components are expressed in a variety of immune cells. In this new scenario, further clinical research investigating the combined effect of RAAS with anti-cancer therapy could represent a rich source of opportunities for combating two of humanity’s most dreaded diseases.

## 6. Conclusions

In recent years, life expectancy has increased notably as technological advances and the obtaining of new knowledge about the physiological and pathological aspects of various diseases including MI. Invasive and pharmacological interventions have allowed an increase in longevity, giving way to other complex diseases such as HF and cancer. Despite a large amount of information, the mechanisms involved in this association are not yet fully understood. There are therapeutic or pharmacological tools that make it possible to reduce the high mortality rates they cause globally. Cancer, which shares a large number of cardiovascular risk factors with other diseases, per se, already represents a substantial cause of mortality. Now we add HF to the picture as a risk factor for its development, the expectations for the future are grim. The use of drugs that target the RAAS has shown promising results in the prevention of secondary cardiovascular events in both MI and HF. These include a lower risk of certain cancers in these patients, as well as current therapies of cancer individually or in combination with other drugs. The use of ACEI and ARBs is emerging to treat inflammatory diseases. Therefore, understanding the mechanisms involved in this complex relationship will provide tools for a better diagnosis and treatment and to improve the prognosis and quality of life of people suffering from these two life-threatening diseases.

## Figures and Tables

**Figure 1 ijms-22-07106-f001:**
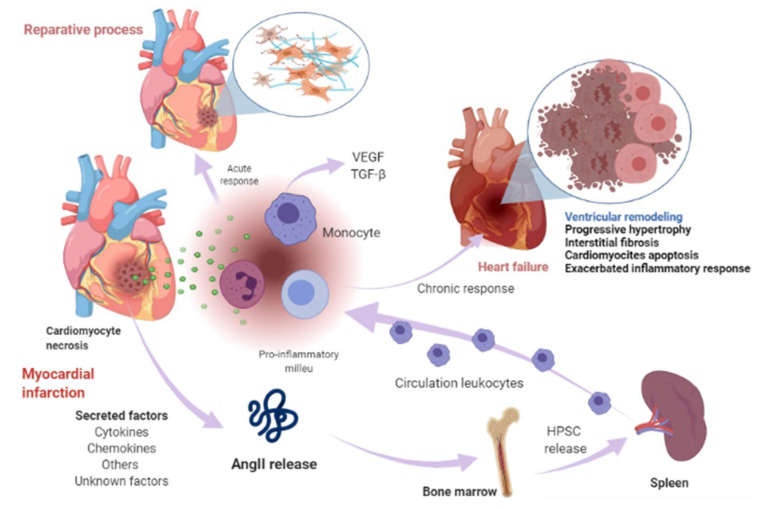
Myocardial infarction and heart-failure-related events. Shortly after myocardial injury, an increase in AngII concentration occurs and induces an accumulation, differentiation, and exit of hematopoietic stem/precursor cells (HPSC) from the bone marrow to contribute to splenic myelopoiesis to supply the infarcted area of the immune cells. Cardiomyocyte necrosis releases signals of danger and induces the secretion of cytokines, chemokines, and adhesion molecules to allow the recruitment and infiltration of leukocytes (mainly monocytes) into the infarcted area. Monocytes exert a reparative response, phagocytosing the cellular debris, while it stimulates repair pathways by secreting pro-inflammatory cytokines through the binding of angiotensin-II (AngII) to type 1 angiotensin-II receptor (AT1R), which induces the phosphorylation of nuclear factor-kappa B (NF-kB). This induces a pro-inflammatory response mediated by tumor necrosis factor-alpha (TNF-α) or interleukin-1 beta (IL1ß) and drives inflammation. The modulation of inflammation in this repair phase includes fibroblast activation and healing mediated in part by renin-angiotensin- aldosterone system (RAAS). When this response becomes chronic, it leads to a pathological process called ventricular remodeling, characterized by progressive hypertrophy of myocytes and interstitial fibrosis, which in later stages involve progressive loss of myocytes through apoptosis, an exacerbated inflammatory response. The healing and the adverse remodeling of the infarcted ventricle ultimately underlie heart failure. This environment can lead to the secretion of certain factors into the circulation that are synthesized in various cell types in the heart, including cardiomyocytes, fibroblasts, smooth muscle, and vascular endothelial cells and other unknown factors. Image created with BioRender.com (Toronto, ON, Canada).

**Figure 2 ijms-22-07106-f002:**
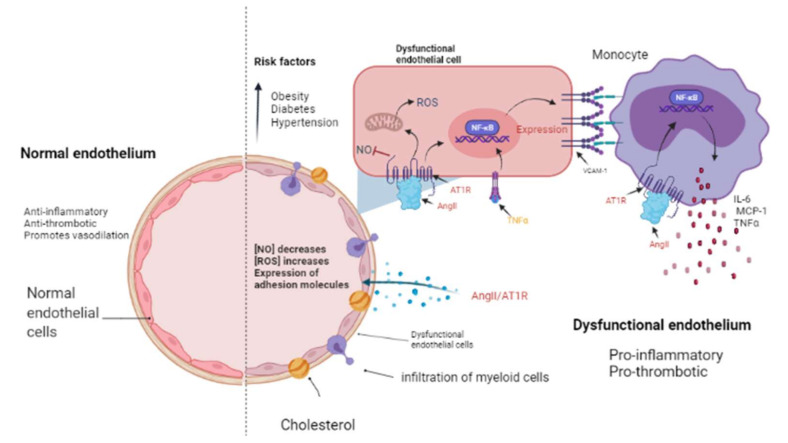
The endothelium in a normal and pathological state. The endothelium is a monolayer of cells that covers the interior of each major and minor vessel. A healthy endothelium has anti-inflammatory, anti-thrombotic properties and promotes vasodilation through nitric oxide (NO) release (**left side**). Cardiovascular risk factors such as obesity, diabetes and hypertension could promote a dysfunctional endothelium (**right side**) that is characterized by a decrease in NO release as well as an increment in reactive oxygen species (ROS) and a pro-inflammatory activity mediated by AngII/AT1R signaling, which activates NF-κB and, consequently, the expression of cytokines, chemokines, and adhesion molecules: interleukin-6 (IL-6), monocyte chemoattractant protein-1 (MCP-1), and vascular cell adhesion molecule-1 (VCAM-1) by endothelial cells. Then, myeloid cells such as monocytes migrate and infiltrate towards the aortic walls (where they become macrophages) contributing to the endothelial dysfunction by producing tumor necrosis factor-alpha (TNF-α), IL-6, and MCP-1. Image created with BioRender.com, Toronto, ON, Canada.

**Figure 3 ijms-22-07106-f003:**
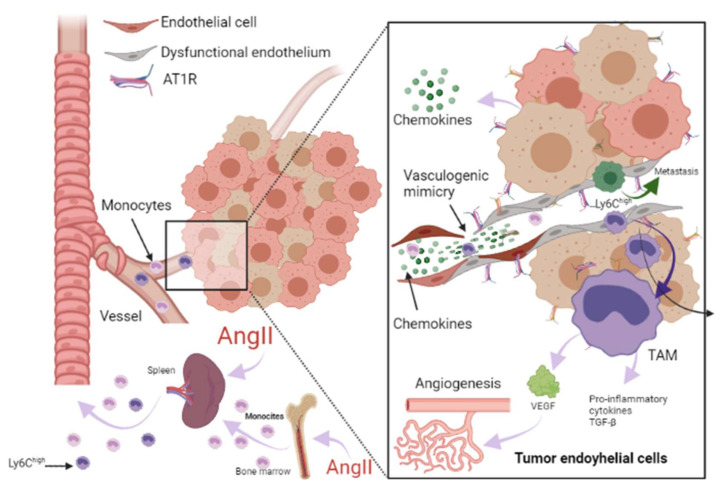
Role of RAAS in the tumor microenvironment. Immune cells can infiltrate tumors and differentiate into tumor-associated macrophages (TAM) derived mainly from circulating monocytes and are attracted to the tumor by chemokines. TAMs can stimulate tumor cell proliferation, angiogenesis, invasion and metastasis. Additionally, tumor microenvironment (TME) can influence the phenotype of circulating monocytes such as the Ly6C^high^ monocyte subset, giving them an immunosuppressive activity and a decreased responsiveness to inflammatory stimuli before their infiltration intoTME. Angiotensin-II (AngII) plays a relevant role in macrophage-mediated chronic inflammation, increasing macrophage progenitors and supplying of TAMs. Additionally, endothelial cells (EC) can promote pro-inflammatory signaling, favoring spontaneous metastasis in adjacent tumors through an aberrant expression of pro-inflammatory cytokines, extracellular matrix, alterations in the leukocyte adhesion process, increasing vascular cell adhesion molecule-1 (VCAM-1) and abnormal responses to oxidative stress. In ECs, AngII/AT1R signaling generates a pro-angiogenic response mediated by vascular endothelial growth factor (VEGF). AngII signaling activates TNF-α and NF-κB and upregulates pro-inflammatory endothelial chemokines. AngII has been reported to be able to promote VCAM-1 expression and enhance adhesion, growth, angiogenesis, and the inflammatory microenvironment through AT1R in hepatocellular carcinoma. It has been reported that angiogenesis promotes tumor cell metastasis. Image created with BioRender.com, Toronto, ON, Canada.

**Table 1 ijms-22-07106-t001:** Findings of observational studies evaluating incidence of cancer in patients with HF.

Clinical Study/Population	Type of Malignancy (Percentage)	HR, 95% IC, *p* Value	Findings
Hasin et al. 2013 [6]. Subjects newly diagnosed with HF and without cancer at index	Digestive system (19%) Male reproductive (19%) Hematologic (16%) Breast ^a^ (10%) Respiratory (8%) Urinary (8%) Female reproductive (3%) Skin (3%) Other cancers ^c^ (14%)	*Not provided*	HF patients had a 68% higher risk of developing cancer (HR: 1.68; 95% CI: 1.13–2.50) adjusted for body mass index, smoking, and comorbidities in the HF group.
Banke et al. 2016 [4]. HF patients without a prior diagnosis of cancer.	Lung (15.7%)	1.81, 1.54–2.12, *p* < 0.0001	Risk of any type of cancer increased (IRR: 1.24; 95% CI: 1.15–1.33, c < 0.0001), except for prostate cancer.
	Skin (16.3%)	1.84, 1.57–2.15, *p* < 0.0001	
	Kidney and urinary system (8.2%)	1.75, 1.41–2.18, *p* < 0.0001	
	Liver/biliary system (4.7%)	1.60, 1.20–2.13, *p* = 0.0015	
	Lymph/blood (6.8%)	1.45, 1.14–1.85, *p* = 0.0027	
	Colon/rectal (12.3%)	1.24, 1.04–1.49, *p* = 0.0180	
	Breast ^a^ (4.8%)	1.36, 1.02–1.81, *p* < 0.038	
	Prostate _b_ (13%) Other ^c^ (18%)	1.04, 0.88–1.24, *p* < 0.6345-	
Hasin et al. 2016 [7]. Survivors of a first MI who developed HF	Respiratory system: 29% Digestive system: 29% Hematologic: 14% Skin: 7% Male reproductive: 4% Breast: 4% Urinary: 4% Female reproductive: 4%	*Not provided*	Patients who develop HF after MI have an increased risk of cancer (HR: 2.16, 95% CI: 1.39–3.35).
Sakamoto et al. 2017 [8]. Chronic HF patients without a prior diagnosis of cancer.	Stomach (0.41%)	95% IC: 0.25–0.61, *p* < 0.0001 *	The incidence of cancer in chronic HF patients was approximately four times higher contrasting with control patients (2.27% vs. 0.59%, 95% CI: 1.89–2.71, *p* < 0.0001).
	Lung (0.22%)	0.12–0.40, *p* = 0.0002	
	Prostate (0.24%)	0.09–0.46, *p* < 0.0001	
	Breast (0.51%)	0.26–0.93, *p* < 0.0001	
	Colon (0.21%)	0.10–0.38, *p* = 0.006	
	Others ^d^	1.89–2.71, *p* < 0.0001	
	Overall cancer (2,27%)		
Kwak et al. 2021 [5]. Patients with HF and aged ≥20 years	Gastrointestinal (3.3%)	(1.49, 1.44–1.54, *p* < 0.0001)	Patients with HF presented a higher risk for cancer development compared to controls (HR: 1.64, 95% CI: 1.61–1.68) and the increased risk was consistent for all site-specific cancers.
	Liver/Biliary/Pancreas (2.2%)	(1.80, 1.72–1.88), *p* < 0.0001)	
	Lung (2%)	(2.22, 2.12–2.32, p< 0.0001)	
	Prostate ^c^ (1.7%)	(1.40, 1.31–1.49, *p* < 0.0001)	
	Hematology (0.7%)	(2.77, 2.55–3.00, *p* < 0.0001)	
	Genitourinary (0.6%)	(1.55, 1.43–1.69, *p* < 0.0001)	
	Thyroid (0.4%)	1.30, 1.18–1.43, *p* < 0.0001)	
	Breast ^b^ (0.6%)	(1.36, 1.21–1.52, *p* < 0.0001)	
	Female reproductive (0.6%)	(1.90, 1.68–2.15, *p* < 0.0001)	
	Head and neck (0.2%)	1.62, 1.41–1.87, *p* < 0.0001	
	Skin (0.04%)	(1.53, 1.11–2.11, *p* = 0.0081)	
	Overall cancer (9.2%)	(1.64, 1.61–1.68, *p* < 0.0001)	

^a^ Only in women, ^b^ Only in men, ^c^ Not specified. ^d^ Other cancers are cervical cancer, sigmoid colon cancer, renal cancer, uterine body cancer, liver cancer, esophageal cancer, rectal cancer, thyroid cancer, bile duct cancer and bladder cancer. Abbreviations: HR: hazard ratio; IRR: incidence rate ratio; IC: confidence interval; MI: myocardial infarction; HF: heart failure. * Variance analysis described in the original paper.

**Table 2 ijms-22-07106-t002:** Role of RAAS inhibitors in heart failure.

RAAS Inhibitor	Observations
**Angiotensin converting enzyme inhibitors (ACEI)**	
Captopril	Long-term administration was associated with an improvement in survival and reduced morbidity and mortality due to major cardiovascular events in patients with asymptomatic left ventricular (LV) dysfunction after myocardial infarction (MI) [82].
Enalapril	Increased exercise time and left ventricular ejection fraction (LVEF) [83].
Perindopril	Increased 6 min walk distance but did not decrease mortality [84]. After 1-year treatment reduced progressive LV remodeling but it was not associated with better clinical outcomes [85].
Ramipril	Administration to patients with clinical evidence of either transient or ongoing heart failure (HF) after MI resulted in a substantial reduction in premature death from all causes [86].
Trandolapril	Long-term treatment in patients with reduced LV function soon after MI significantly reduced the risk of overall mortality, mortality from cardiovascular causes, sudden death, and the development of severe HF [87].
**Angiotensin II type 1 receptor blockers (ARBs)**	
Telmisartan	Telmisartan was well tolerated in patients unable to tolerate ACEI. Although the drug had no significant effect on hospitalizations for HF, it modestly reduced the risk of the composite outcome of cardiovascular death, MI, or stroke [88].
Candesartan	Slightly decreased hospitalizations but did not decrease mortality [89]. Reduced cardiovascular mortality and hospital admissions for worsening chronic HF. Patients with reduced ejection fraction were the most benefited [90].
Losartan	Reduced the rate of death or admission for HF in patients with HF, reduced LVEF, and intolerance to ACEI [91].
Valsartan	In patients with MI associated with HF and/or LV dysfunction, valsartan administration in the immediate post MI period demonstrated equal efficacy than captopril [92,93].
**Aldosterone antagonists**	
Spironolactone	Prevented LV fibrosis and remodeling after MI [94]

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
