# Peer review of "RAAS: A Convergent Player in Ischemic Heart Failure and Cancer"

_ijms, 2021, doi:10.3390/ijms22137106_

Round 1
Reviewer 1 Report
- Authors introduced an intriguing vision on cancer as a trigger of heart failure. Yet, authors did not introduce chemotherapy as a cause of cardiotoxicity that leads to cardiac dysfunction (cardiomyopathy) and eventually to heart failure. Impact of “pathophysiology of cancer” and “cardiotoxicity” resulted from cancer therapies’ should be clearly distinguished.
- Misleading representation of heart failure in Introduction:
Lines 38-40: “The American College of Cardiology defines heart failure as a complex clinical syndrome that results from any structural or functional alteration of ventricular filling or blood ejection (2).” – Author omitted dyspnea, fluid retention/congestion from HF definition described in ref. 2. Such selective quotation makes HF be mistakenly presented/introduced as a cardiac dysfunction (cardiomyopathy).
Here is the complete citation related to HF definition from Ref. 2: “HF is a complex clinical syndrome that results from any structural or functional impairment of ventricular filling or ejection of blood. The cardinal manifestations of HF are dyspnea and fatigue, which may limit exercise tolerance, and fluid retention, which may lead to pulmonary and/ or splanchnic congestion and/or peripheral edema.”
Lines 40-43: “Heart failure is a heterogeneous syndrome (3) it has an estimated 17 primary etiologies (4), however, about 80% of all HF cases can be attributed to two diseases principally: ischemic heart disease (42.3%) and chronic obstructive pulmonary disease (37.0%) (5).” – COPD is comorbidity rather than cause of HF. Major cause of HF are coronary artery disease, ischemic disease and cardiomyopathies.
- Authors refers to renin-angiotensin-aldosterone system (RAAS) outdated way calling it RAS. Aldosterone is established important member of RAAS and one of the major player of RAAS system causing fluid and water retention (edema/congestion) that defines heart failure from heart dysfunction.
Lines 74-76: “A hallmark of heart failure is chronic activation of the renin-angiotensin system (RAS), especially marked by a systemic increase in levels of angiotensin-II (AngII), the effector peptide of this neurohumoral system “ – Ref (13) related to mouse model does not sufficiently support this statement. Many solid original publications and comprehensive reviews are available to support chronic RAAS activation as a hallmark of HF
Introduction and review sub-sections in general are unfocused.
- Formatting issues:
“Heart failure” (Lines 46, 52 etc.) and “HF” (Lines 89, 199, 120) are randomly used over the text.
References’ information is not fully provided (4; 5; 9; 10; 12; 15; 20; 21; etc.).
Ref 13, line 423 – word “angiotensin” is missing.
Author Response
Dear reviewer.
We are submitting the revised version IJMS 11880818 by García-Garduño TC et al. We appreciate the opportunity of improving the manuscript. All comments were taken into account. The changes are highlighted as “Track changes” as it was suggested for an easy identification. Please see the attachment.
Best regards.
Yeminia Valle, Corresponding author
Instituto de Investigación en Ciencias Biomédicas, Centro Universitario de Ciencias de la Salud, Universidad de Guadalajara.

Reviewer 2 Report
Based on the pathophysiological mechanisms of heart failure and myocardial infarction, this study reviewed the potential connection of the two disease categories, HF and cancer. Some major mediators and their preclinical and clinical implications were thoroughly reviewed. Pathophysiological functions of HF-related neurohormonal responses and their role in the tumor microenvironment were summarized, addressing the results of diverse previous investigations. Since this topic has not been commonly applied in clinical oncology, the current review can provide some novel points of view in the contemporary real-world clinics.
Here are some suggestions. Clinical investigations are relatively insufficient in this topic. If the clinical data the authors reviewed here were categorized according to the type of malignancy (as a table, if possible), such as breast and lung cancer, readers can easily recognize the level of clinical evidence regarding this point of view. With increasing interests about cancer immunology these days, some comments on the clinical implications especially in the era of cancer immunotherapy would be informative. In addition, the length of some sentences is relatively long and the overall manuscript is not easily readable. I recommend additional editing process to improve readability.
Author Response
Dear reviewer.
We are submitting the revised version IJMS 11880818 by García-Garduño TC et al. We appreciate the opportunity of improving the manuscript; all comments were taken into account. The changes are highlighted as “Track changes” as it was suggested for an easy identification. Finally, we want to inform you that a native English-speaking colleague checked the manuscript. Please see the attachment.
Best regards.
Yeminia Valle, Corresponding author
Instituto de Investigación en Ciencias Biomédicas, Centro Universitario de Ciencias de la Salud, Universidad de Guadalajara.

Reviewer 3 Report
The authors provide an overview of the mechanisms linking heart failure (HF) to cancer, with emphasis on the renin-angiotensin system (RAS).
I have the following remarks:
1) the title is misleading, it should clearly indicate that the review focuses on the role of the RAS in the interconnection between HF and cancer
2) the Introduction is too long and largely consists of very general paragraphs, e.g. those reporting the definitions of HF and cancer and the hallmarks of cancer biology. By contrast, it could include a section taking into account the other possible explanations for the emerging association of HF with cancer, in particular the improved survival of HF patients - see J Am Heart Assoc. 2020 Sep 15;9(18):e016309 and Eur J Clin Invest. 2020 Sep;50(9):e13273
3) another important difference between the article by Selvaraj et al. and the other ones investigating the association of HF with cancer is that HF was self-defined by the study participants, instead of being assigned based on ICD-9 codes and registry data
4) the reader might be hepled in following the text by a couple of Figures, which could display some of the concepts put together in the Graphical Abstract
5) legend to the Graphical Abstract: the authors should state that, at present, the connections between HF and cancer are hypothesized
6) Table 1 is long and confusing, mixing together clinical and experimental studies. I suggest breaking it into two tables, one summarizing the preclinical findings and another summarizing the clinical ones. Furthermore, the text should be reduced
Author Response
Dear reviewer.
We are submitting the revised version IJMS 11880818 by García-Garduño TC et al. We appreciate the opportunity of improving the manuscript; all comments were taken into account. The changes are highlighted as “Track changes” as it was suggested for an easy identification. Finally, We want to inform you that a native English-speaking colleague checked the manuscript. Please see the attachment.
Best regards.
Yeminia Valle, Corresponding author
Instituto de Investigación en Ciencias Biomédicas, Centro Universitario de Ciencias de la Salud, Universidad de Guadalajara.

Reviewer 4 Report
The manuscript is well done and can be published in form!
Author Response
Dear reviewer.
We appreciate the time you spent reading the manuscript. We are submitting the revised version IJMS 11880818 by García-Garduño TC et al. Finally, we want to inform you that a native English-speaking colleague checked the manuscript.
Best regards.
Yeminia Valle, Corresponding author
Instituto de Investigación en Ciencias Biomédicas, Centro Universitario de Ciencias de la Salud, Universidad de Guadalajara.
Round 2
Reviewer 1 Report
The author team needs to invite an expert/experts in heart failure/or cardiovascular disease to cover heart failure-related topics and the accuracy of provided statements.
Below just a few misleading statements from the Introduction or statements that require support from the appropriate references:
Lines 38-39: Misleading statement and not-connected reference: “Nevertheless, HF is caused more frequently by a cardiac injury related to myocardial infarction (MI) [4].”
- Ischemic heart disease or coronary heart disease(CHD)/coronary artery disease is the most frequent HF causation. Myocardial infarction (MI) is the only one of many manifestations of ischemic heart disease.
- Reference #4 is related to ischemic heart disease and mentioned the heart failure only once. Appropriate references are required.
- While ischemic heart disease is the most frequent cause of HF with known ethology, other cardiac pathologies such as hypertension, cardiomyopathies, valves pathologies, congenital heart disease, arrhythmias, myocarditis, also significantly contribute to heart failure development.
- Lines 40-42: “In recent years, various epidemiological studies have reported the contrary scenario, patients with HF have an increased risk of cancer compared to general population and this new connection represents a challenge for the field of cardio-oncology.” – this statement should be supported by publications, otherwise it is opinion statement rather than fact-based and widely approved statement.
The same concern is related to Line 154: “Myocardial infarction is the most common cause of HF globally [4].”
Lines 50-53: “Neurohumoral response includes the renin-angiotensin-aldosterone system (RAAS) which can be chronically stimulated representing a hallmark of HF and also it has been established that RAAS is frequently altered in a variety of cancer types, associated with a poor prognosis [7]”
While Reference #7 strongly supports the role of RAAS in cancer, it does not mention “heart failure even ones”.
Hence, the statement requires support from other publications connecting RAAS and HF. Just from the quick PubMed search, there are plenty of classical and recent reviews:
- Schrier, R.W.; Abraham, W.T. Hormones and hemodynamics in heart failure. Engl. J. Med.1999, 341, 577–585.
- Weber, K.T. Aldosterone in congestive heart failure. Engl. J. Med.2001, 345, 1689–1697.
- Patel, V.B.; Zhong, J.C.; Grant, M.B.; Oudit, G.Y. Role of the ACE2/Angiotensin 1-7 Axis of the Renin-Angiotensin System in Heart Failure. Res.2016, 118, 1313–1326.
- Hartupee, J.; Mann, D.L. Neurohormonal activation in heart failure with reduced ejection fraction. Rev. Cardiol.2017, 14, 30–38.
- Orsborne C, Chaggar PS, Shaw SM, Williams SG. The renin-angiotensin-aldosterone system in heart failure for the non-specialist: the past, the present, and the future.
Postgrad Med J. 2017 Jan;93(1095):29-37.
- Sullivan, R.D.; Mehta, R.M.; Tripathi, R.; Reed, G.L.; Gladysheva, I.P. Renin Activity in Heart Failure with Reduced Systolic Function-New Insights. J. Mol. Sci. 2019, 20, 3182.
- Ocaranza, M.P.; Riquelme, J.A.; Garcia, L.; Jalil, J.E.; Chiong, M.; Santos, R.A.S.; Lavandero, S. Counter-regulatory renin-angiotensin system in cardiovascular disease. Rev. Cardiol.2020, 17, 116–129.
- Pugliese NR, Masi S, Taddei S.The renin-angiotensin-aldosterone system: a crossroad from arterial hypertension to heart failure.Heart Fail Rev. 2020 Jan;25(1):31-42.
With respect to RAAS:
Line 140: the misleading statement “The classical or “circulating” RAAS consists of several components.”
Classical RAAS refers to renin-AngII/ACE/AT1/AT2-aldosterone axis of RAAS in contrast to alternative RAAS – ACE2/Ang(1-7)/MAS that aims to compensate classical RAAS. Both systems could be systemic or tissue-related.
The current review completely overlooks the role of alternative RAAS axis ACE2/Ang(1-7)/MAS in HF and cancer.
Author Response
Dear Reviewer
We appreciate all your valuable comments, all of them were carefully addressed.
Please see the attachment.
Kind regards.

Reviewer 3 Report
The authors addressed all the remarks I made
Author Response
Dear reviewer
Thank you for your valuable time.
Kind regards.
Round 3
Reviewer 1 Report
The authors precisely overview RAAS as one of the sharing mechanisms between HF of ischemic origin and cancer. Since authors specifically target only HF of ischemic origin (post-MI) or Ischemic heart failure, omitting HF caused by other cardiac pathologies such as hypertension, cardiomyopathies, valves pathologies, congenital heart disease, arrhythmias, myocarditis, the review title should be consistently adjusted to “RAAS: a convergent player in Ischemic heart failure and Cancer.” The current title “RAAS: a convergent player in Heart failure and Cancer” sounds too broad, and may mislead readers in their expectations.
Author Response
We agree with this suggestion.
We appreciate the valuable contributions you have made to improve this manuscript.
Kind regards.
Yeminia Valle.
Corresponding author.
Universidad de Guadalajara.